# AUTO DP-SGD: DUAL IMPROVEMENTS OF PRIVACY AND ACCURACY VIA AUTOMATIC CLIPPING THRESHOLD AND NOISE MULTIPLIER ESTIMATION

## ABSTRACT

Differentially Private Stochastic Gradient Descent (DP-SGD) has emerged as a popular method to protect personally identifiable information (PII) in deep learning (DL) applications. Unfortunately, DP-SGD's per-sample gradient clipping and uniform noise addition during training can significantly degrade model utility. To enhance the model's utility, researchers proposed various adaptive/dynamic DP-SGD methods by adapting the noise multiplier and clipping threshold. However, we examine and discover that these established techniques result in greater privacy leakage or lower accuracy than the traditional DP-SGD method, or a lack of evaluation on a complex data set such as CIFAR100. To address these limitations, we propose an automatic DP-SGD (Auto DP-SGD). Our method automates clipping threshold estimation based on the DL model's gradient norm and scales the gradients of each training sample without losing gradient information or requiring an additional privacy budget than what is needed for DP training. This helps to improve the algorithm's utility while using a less privacy budget. To further improve accuracy, we introduce automatic noise multiplier decay mechanisms to decrease the noise multiplier after every epoch. Finally, we develop closed-form mathematical expressions using the truncated concentrated differential privacy (tCDP) accountant, which offers a straightforward and tight privacy-bound analysis for automatic noise multiplier and automatic clipping threshold estimation. Through extensive experimentation, we demonstrate that Auto DP-SGD outperforms existing state-of-the-art (SOTA) DP-SGD methods in privacy and accuracy on various benchmark datasets. We also show that privacy can be improved by lowering the scale factor and using learning rate schedulers without significantly reducing accuracy. Moreover, we explain how to select the best Auto DP-SGD variant that does not require a privacy budget more than what is needed to train the Auto DP algorithm. Specifically, Auto DP-SGD, when used with a step noise multiplier (Auto DP-SGD-S), improves accuracy by 3.20%, 1.57%, 6.73%, and 1.42% for the MNIST, CIFAR10, CIFAR100, and AG News Corpus datasets, respectively. Furthermore, it obtains a substantial reduction in the privacy budget ($\epsilon$) of 94.9%, 79.16%, 67.36%, and 53.37% for the corresponding data sets.

## 1 INTRODUCTION

DL emerged as a powerful technology during the fourth industrial revolution Sarker (2021). Business intelligence, sentiment analysis, banking, healthcare Ardila et al. (2019), finance Huang et al. (2020), and many other fields employ DL to earn huge revenue and reduce human burden hea (2023); Fin (2023). Unfortunately, data, including patient images and PII Hassani et al. (2020), used to train DL algorithms in some industries mentioned above, are privacy sensitive. Current studies demonstrate that extracting sensitive information from DL models is possible through various attacks Shokri et al. (2017); Hu et al. (2022); Truex et al. (2018); Gong & Liu (2016; 2018); Zhao et al. (2021); Fredrikson et al. (2015); Wu et al. (2016); Chen et al. (2020a); Dwork et al. (2014); Dinur & Nissim (2003). Even more concerning, sensitive information cannot be protected using conventional methods like de-identification Near & Abuah (2021) and $k$-anonymity Sweeney (2015).

The existing investigation indicates that differential privacy (DP) can provide strong privacy guarantees for sensitive information Dwork et al. (2011; 2006b). Different DP mechanisms are implemented in various machine learning (ML) algorithms Fletcher & Islam (2019); Jagannathan et al. (2009); Chaudhuri et al. (2011); Wang et al. (2017), and deep neural networks (DNN) Abadi et al. (2016); Zhang et al. (2021) For DNN, DP-SGD Abadi et al. (2016); Chen et al. (2020b) is used more frequently as it obtains higher accuracy with reasonable privacy loss. DP-SGD uses per-sample gradient clipping to bind the sensitivity of the gradients and adds noise to the aggregated clipped gradients to avoid leakage of PII. Using uniform clipping Abadi et al. (2016); Koskela & Honkela (2020); Zhang et al. (2021) can result in increased privacy leakage, particularly when using elevated clipping thresholds such as 50 as shown in Table 2. Du et al. (2021) proposed dynamic DP-SGD incorporating dynamic clipping and adaptive noise addition. However, they did not adequately estimate the privacy budget and did not perform experiments on complex data sets such as CIFAR10 and CIFAR100. We implement dynamic DP-SGD on the CIFAR-10 data set and find it to have lower accuracy than conventional DP-SGD, as shown in Table 3.

To estimate the privacy budget of the DP algorithms, there are different privacy accountants such as concentrated DP (CDP) Dwork & Rothblum (2016), zero concentrated DP (zCDP) Bun & Steinke (2016), Renyi DP (RDP) Mironov (2017) and truncated concentrated DP (tCDP). CDP and zCDP accountants lack privacy amplification by sampling. The RDP accountant Abadi et al. (2016), overestimates privacy costs. On the other hand, the Gaussian DP (GDP) accountant Du et al. (2021) underestimates the exact privacy budget. Auto DP-SGD adjusts the noise multiplier and clipping threshold after each epoch and sample, respectively. The privacy accountant of tCDP provides lemmas 1-4 to incorporate these changes, support privacy amplification through subsampling, and develop closed-form mathematical expressions that efficiently approximate the total privacy budget, as opposed to CDP, zCDP, RDP, and GDP.

In conclusion, existing work suffers from issues such as the poor choice of privacy accountant Du et al. (2021), uniform clipping Abadi et al. (2016); Zhang et al. (2021); Koskela & Honkela (2020), and limited evaluation Du et al. (2021). To overcome these limitations and bridge the gap between DP and nonprivate model accuracy while maintaining strong privacy guarantees, we design Auto DP-SGD. The Auto DP-SGD technique comprises two crucial components: an automatic clipping threshold estimation and an automatic noise multiplier estimation. Automatic clipping threshold estimation avoids operations involving a batch of model parameters or gradients that are obtained over a batch of data samples, as they compromise differential privacy. Our method uses the total gradient norm of the model for a single sample in a batch to determine the clipping threshold for that sample. This approach offers two advantages: First, it lowers the average sensitivity of the algorithm by adjusting the clipping threshold according to the decreasing gradient norm throughout the training; thus, it improves privacy; second, it eliminates excess noise, leading to accuracy improvement. Inspired by the usage of learning rate schedulers in non-private deep learning models, we formulate the concept of an automatic noise multiplier decay for DP-SGD. Our work introduces the following primary contributions:

- To accurately estimate the clipping threshold for each sample, we propose an algorithm 1 that computes the total gradient norm of the model, multiplies it with a scale factor and sets it as a clipping threshold. Then, it scales all the per-sample gradients and avoids clipping. Scaling the gradients has been shown to help the model converge faster, and thus it improves the utility of the model Bu et al. (2021); Esipova et al. (2022).
- We propose three new automatic noise multiplier decay mechanisms: (i) time decay, (ii) exponential decay, and (iii) step decay by extending the existing linearly decaying noise multiplier Zhang et al. (2021). Decay of the noise multiplier reduces the noise multiplier after every epoch and helps to improve the accuracy of the model Zhang et al. (2021). Our Auto DP-SGD integrates an automatic noise multiplier decay mechanism and automatic clipping threshold estimation algorithm.
- We investigate the impact of various learning rate schedulers, scale factors, and noise multiplier decay mechanisms on DP models' accuracy and privacy. Moreover, we develop mathematical expressions to estimate the privacy budget of Auto DP-SGD.
- Through extensive evaluations, we demonstrate that Auto DP-SGD outperforms current state-of-the-art (SOTA) DP-SGD methods on various benchmark datasets. More specifically, our Auto DP-SGD-S variant improves accuracy by 3.20%, 1.57%, 6.73%, and 1.42% as well as reduces the privacy budget ($\epsilon$) by 94.9%, 79.16%, 67.36%, and 53.37% on the MNIST, CIFAR10, CIFAR100, and AG News Corpus datasets, respectively. Moreover, we

also demonstrate how to select the best Auto DP-SGD-S variant that does not require a privacy budget more than what is needed to train the Auto DP algorithm.

## 2 RELATED WORK

***Adaptive DP-SGD algorithms.*** There are many studies related to DP-SGD. This section only compares the most related work with our proposed Auto DP-SGD. Existing work has focused on developing adaptive hyperparameters such as learning rate, clipping threshold, and noise multiplier to improve the trade-off between privacy and accuracy. Zhang et al. (2021) proposed adaptive DP-SGD that linearly decays the Gaussian noise mechanism to reduce noise, demonstrating its better performance than the standard DP-SGD approach. Koskela & Honkela (2020) presented a DP technique that eliminates the need for validation sets when adapting the learning rate for DP-SGD. Du et al. (2021) proposed a dynamic DP-SGD approach that dynamically adjusts the clipping and noise multiplier. These studies change one or two of the three hyperparameters (learning rate, noise multiplier, and clipping threshold) to improve the DP-SGD. In contrast, our work proposes an optimization method that addresses all three parameters simultaneously and obtains SOTA accuracy and privacy.

DP-SGD Global Bu et al. (2021) and DP-SGD Global-Adapt Esipova et al. (2022) are popular gradient scaling algorithms for DP-SGD. DP-SGD Global scales gradients with an $l_2$ norm less than or equal to a clipping threshold. If the gradients are larger than the clipping threshold, they are discarded. DP-SGD Global has two problems: (i) If the clipping threshold is set too high, no gradients are discarded, but the clipped gradients become smaller, making it harder for the algorithm to converge. (ii) If the clipping threshold is too low, most gradients are discarded, leading to information loss. DP-SGD Global-Adapt is a modified version of DP-SGD Global that makes the DP algorithm fairer. It does this by clipping the gradients higher than the upper clipping threshold to have an $l_2$ norm equal to the lower clipping threshold to reduce the loss of information. It also adaptively changes the upper threshold to be higher than all the per-sample gradients by using the privacy budget. Since the fairness of DP is not the focus of our paper, we leave testing our algorithm on a fairness-related dataset to future work. Our Auto DP-SGD algorithm does not require an additional privacy budget than what is needed for the DP algorithm training and does not cause any information loss.

***Privacy accountants.*** Privacy accountants compose the loss of privacy incurred during each iteration of DP training to calculate the total cost of privacy $(\epsilon, \delta)$. Dwork et al. (2006a); Dwork & Lei (2009) offers a simple composition method that linearly combines the DP of various iterations, resulting in a greater loss of privacy. Dwork et al. (2010) defined an advanced composition theorem to tightly bind the cumulative privacy budget. Abadi et al. (2016) proved that tighter estimates of the total privacy loss could be obtained by tracking higher moments of the privacy loss. Mironov (2017) introduced an RDP based on Renyi divergence to track cumulative privacy loss throughout training. RDP underestimates the true cost of privacy. Dong et al. (2019) proposed $f$-DP measure the privacy cost from the point of view of hypothesis testing, with GDP as the main application. However, while GDP permits a tight composition, it is computationally difficult to determine the accurate composition of the Gaussian mechanism with subsampling amplification. Bun et al. (2018) proposed tCDP as an enhancement over CDP. tCDP supports privacy amplification, unlike CDP, and offers a method to increase accuracy exponentially. Recently, Gopi et al. (2021) have proposed numerical methods to determine the optimal composition of the DP mechanisms. It is difficult to calculate how much privacy the algorithm loses when the noise multiplier changes for each epoch and the clipping threshold changes for each sample, as in Auto DP-SGD.

## 3 BACKGROUND ON DIFFERENTIAL PRIVACY (DP)

Differential privacy (DP) Dwork (2008); Hilton (2002) is a method to preserve an individual's data while revealing aggregated information. DP is formally defined as follows:

**Definition 3.1.** *A randomized function $\mathcal{F} : \mathcal{D} \to \mathcal{R}$ with a domain $\mathcal{D}$ and range $\mathcal{R}$ satisfies $(\epsilon, \delta)-$ differential privacy if for any two datasets $d, \hat{d} \in \mathcal{D}$, differing with only a single data sample and for any subset of outputs $O \subseteq \mathcal{R}$, it holds that*

$$Pr[\mathcal{F}(d) \in O] \leq e^\epsilon Pr[\mathcal{F}(\hat{d}) \in O)] + \delta \tag{1}$$

One commonly used method to introduce randomness to a deterministic real-valued function $g$ : $\mathcal{D} \rightarrow \mathcal{R}$ is by adding noise calibrated to the sensitivity $s_g$ of the function $g$. Sensitivity is the maximum absolute difference between the output of $g$ on any two neighboring data sets $d, \hat{d} \in \mathcal{D}$.

$$s_g = max_{d,\hat{d}} ||g(d) - g(\hat{d})||_2 \tag{2}$$

Most commonly, noise is drawn from the Gaussian distribution and added to the deterministic function as follows:

$$\mathcal{F}(d) = g(d) + \mathcal{N}(0, s_g^2 \cdot \sigma^2) \tag{3}$$

where $\mathcal{N}(0, s_g^2 \cdot \sigma^2)$ is the Gaussian distribution with mean 0 and standard deviation $s_g \cdot \sigma$ and $\sigma$ is termed the noise multiplier.

The function $\mathcal{F}$ satisfies $(\epsilon, \delta)-$ DP, where $\delta \in (0,1)$ and $\sigma \geq \frac{\sqrt{2ln(1.25)/\delta}s_g}{\epsilon}$ is a noise multiplier.

**Definition 3.2.** *(tCDP). For all $\tau \in (1, \omega)$, a randomized algorithm $\mathcal{A}$ is $(\rho, \omega)-$ tCDP if for any neighboring data sets $d$ and $\hat{d}$, and all $\alpha > 1$, we have:*

$$D_\tau(\mathcal{A}(d)||\mathcal{A}(\hat{d})) \leq \rho\alpha \tag{4}$$

where $D_\tau(\cdot||\cdot)$ is the Renyi divergence of order $\tau$.

Given two distributions $\mu$ and $\nu$ on a Banach space (Z,$|| \cdot ||$), the Rényi divergence is computed as follows.

**Definition 3.3.** *Rényi divergence Rényi (1961): Let $1 < \alpha < \infty$, and $\mu, \nu$ be measures with $\mu \ll \nu$. The Rényi divergence of orders $\alpha$ between $\mu$ and $\nu$ is defined as:*

$$D_\alpha(\mu||\nu) \doteq \frac{1}{\alpha - 1} ln \int (\frac{\mu(z)}{\nu(z)})^\alpha \nu(z)dz. \tag{5}$$

Here we follow the convention $\frac{0}{0} = 0$. If $\mu \not\ll \nu$, we define the Renyi divergence as $\infty$. The Renyi divergence of orders $\alpha = 1, \infty$ is defined by continuity.

In this work, we mainly use the following properties of tCDP, as demonstrated in Bun et al. (2018):

**Lemma 1.** *The Gaussian mechanism satisfies $(\frac{(s_g^2)}{2\sigma^2}, \infty)-$ tCDP.*

**Lemma 2.** *If randomized functions $\mathcal{F}_1$ and $\mathcal{F}_2$ satisfy $(\rho_1, \omega_1)$-tCDP and $(\rho_2, \omega_2)$-tCDP, their composition defined as $(\mathcal{F}_1 \circ \mathcal{F}_2)$ is $(\rho_1 + \rho_2, min(\omega_1, \omega_2))-$ tCDP.*

**Lemma 3.** *If a randomized function $\mathcal{F}$ satisfies $(\rho, \omega)-$ tCDP, then for any $\delta \geq 1/exp((\omega - 1)^2 \rho)$, $\mathcal{F}$ satisfies $(\rho + 2\sqrt{\rho ln(1/\delta)}, \delta)-$ differential privacy.*

**Lemma 4.** *If a randomized function $\mathcal{F}$ satisfies $(\rho, \omega)-$ tCDP, then for any $n$-element data set $D$, computing on uniformly random $cn$ entries ensures $(13c^2\rho, log(1/c)/(4\rho))-$ tCDP, with $\rho, c \in (0, 0.1]$, $log(1/c) \geq 3\rho(2 + log(1/\rho))$ and $\omega \geq log(1/c)/(2\rho)$.*

The lemma 1 provides a relation between the Gaussian mechanism and the tCDP privacy accountant. Lemma 2 describes the composition property of two randomized functions under tCDP. The lemma 3 provides a way to convert the privacy budget in the tCDP accountant to the standard $(\epsilon, \delta)-$ DP. The lemma 4 illustrates privacy amplification through random sampling using tCDP. We derive the proof for our proposed algorithm using these lemmas as a basis. We have provided the full proof for all of our Auto DP-SGD variants in Appendix A.1.

---

**Algorithm 1:** Automatic Clipping threshold estimation (AC)

---

**Input:** Batch size $B$, scale factor $W$, batch of model gradients $G$, $G$ includes $g_0, g_1,..., g_{B-1}$
**Initialize an empty clip list** $C$
**for** $b = 0, 1, ..., B-1$ **do**

  **Compute the total gradient norm of the model for current iteration**
  $L = ||g_b||_2$ where $g_b$ represents gradients of all the model parameters after $b^{th}$ sample in $G$
    is backpropagated and the model is updated.
  **Compute the current iteration clipping threshold**
  $c_b = W \cdot L$
  **scale the model gradient**
  $\bar{g}_b = g_b/max(1, \frac{||g_b||_2}{c_b}) = W \cdot g_b$
  **Replace the normal gradient with the scaled gradient**
  $g_b = \bar{g}_b$
  **Add the** $c_b$ **to the clip list** $C$ **and update the gradient list** $G$
**end**
**Output:** $s_t = (\sum_{b=0}^{B-1} c_b)/B, \bar{g}_t = (\sum_{b=0}^{B-1} g_b)/B$

---

## 4 METHODOLOGY

### 4.1 AUTOMATIC CLIPPING THRESHOLD ESTIMATION

DP-SGD employs per-sample gradient clipping to estimate the sensitivity of gradients. Seminal work on DP-SGD Abadi et al. (2016) and some recent studies Zhang et al. (2021) adopt a fixed clipping threshold throughout training. However, the magnitudes of gradients tend to decrease over iterations, and gradient clipping becomes less effective when the gradients are smaller than the clipping threshold. Consequently, using a fixed clipping threshold introduces redundant noise to the model, eventually reducing accuracy. Thus, it becomes necessary to dynamically adjust the clipping threshold between iterations. To address this issue, we propose Algorithm 1.

The algorithm 1 takes the batch size $B$, the scale factor $W(W \in (0, 1])$, and a batch of model gradients $G$ consisting of $g_0, g_1, ..., g_{B-1}$ as inputs. Next, it initializes an empty clip list $C$ to store the clipping threshold $c_b$ for each sample. As the algorithm traverses each sample in the training set, it computes the total gradient norm ($L$) of the model gradients. The $L$ is defined as the $l_2$ norm of the gradients of the model in the current iteration. The clipping threshold ($c_b$) is then calculated by multiplying $L$ by the scale factor $W$. The algorithm then scales the model gradient by computing $\bar{g}_b = W \cdot g_b$. Then it replaces the normal gradient with a scaled gradient. Next, $c_b$ is added to the list $C$, and the gradient list $G$ is updated with $\bar{g}_b$. After running through all iterations, the algorithm produces the average of the clipping threshold and the gradients, respectively. In general, the algorithm 1 scales the gradients of the model and computes the clipping threshold in each iteration. One of the outputs of the algorithm 1, which is the average clipping threshold, is used when estimating the total privacy budget of the Auto DP-SGD. We emphasize that this is the main reason we propose the algorithm 1. We avoid clipping the gradients, since it results in information loss, and scaling helps the model to converge faster, thus improving the utility Bu et al. (2021); Esipova et al. (2022). Algorithm 1 is compatible with DP since computations use per-sample gradients only. We demonstrate the importance of automatic clipping in Appendix C. and the estimation of the total privacy budget in Appendix A.

Table 1: Types of noise multiplier decay mechanisms.

| Auto DP-SGD variant | Decay type | Mathematical expression |
|---|---|---|
| Auto DP-SGD-L | Linear decay Zhang et al. (2021) | $\sigma_e^2 = \sigma_0^2 R^e, R = 0.99$ |
| Auto DP-SGD-T | Time decay | $\sigma_e^2 = \frac{\sigma_0^2}{1+Re}, R = \sigma_0/E$ |
| Auto DP-SGD-S | Step decay | $\sigma_e^2 = \sigma_0^2 R^{\lfloor e/D \rfloor}, R = 0.5, D = 10$ |
| Auto DP-SGD-E | Exponential decay | $\sigma_e^2 = \sigma_0^2 \exp(-Re), R = 0.1$ |

---

**Algorithm 2:** Automatic differentially private SGD (Auto DP-SGD)

---

**Input:** Private examples $\{x_1, ..., x_M\}$, loss function $\mathcal{L}(\theta_t) = \frac{1}{M} \sum_i \mathcal{L}(\theta_t, x_i)$, $\theta_t$ is model
   parameter at $t^{th}$ iteration, noise multiplier decay mechanism $\sigma_e^2 = F(e, R, D, \sigma_0)$,
   learning rate $\eta_t$, Batch Size $B$, $B_t$ is batch of samples at $t^{th}$ iteration, scale factor $W$,
   decay rate $R$, sampling rate $q = B/M$, epochs $E$, iterations $T = E/q$.
**Initialize** $\theta_0$, $\sigma_0$
**for** $t = 0, 1, ..., T - 1$ **do**
$\quad$ **Initialize** an empty list $G$
$\quad$ Randomly take a batch of data samples with sampling rate $q$ from the training data set.
$\quad$ **Compute gradient**
$\quad$ For each $x_i \in B_t$, compute $g = \bigtriangledown_{\theta_t} \mathcal{L}(\theta_t, x_i)$ and add it to $G$
$\quad$ **Call the automatic clipping algorithm to obtain the average of clipping threshold and**
$\quad$ **gradients**
$\quad$ $s_t, \bar{g}_t = AC(B, W, G)$
$\quad$ **Calculate the noise multiplier**
$\quad$ $\sigma_t^2 = F(e, R, D, \sigma_0)$, $e = \lfloor qt \rfloor$
$\quad$ **Add Gaussian noise according to the noise decay scheduler**
$\quad$ $\bar{g}_t = \bar{g}_t + \frac{\mathcal{N}(0, \sigma_t^2 s_t^2 \mathbf{I})}{B}$, where $\mathbf{I}$ is the identity matrix.
$\quad$ **Update the model parameter**
$\quad$ $\theta_{t+1} \leftarrow \theta_t - \eta_t \bar{g}_t$
**end**
**Output:** $\theta_T$

---

## 4.2 AUTOMATIC NOISE MULTIPLIER ESTIMATION

As the iterations progress, the gradients decrease, as shown in Figure 2 for DP-SGD Abadi et al. (2016). However, the noise multiplier is the same across the training. In that case, it is possible that the noise will overpower the gradients, especially in later training iterations when the gradients are much smaller, leading to meaningless model predictions. Moreover, it is necessary to decrease the noise multiplier through training to improve the utility Zhang et al. (2021). Therefore, Zhang et al. (2021) used a linear Gaussian decay noise multiplier to minimize the negative impact of fixed noise addition. We build upon their work and propose three more decaying mechanisms: step decay, exponential decay (exp decay), and time decay inspired by learning rate schedulers used in non-private settings. The various noise multiplier decay techniques examined in this paper are illustrated in Table 1. In Table 1, $\sigma_0$ is the initial noise multiplier; $e$ is the epoch number; $R$ is the decay rate; $D$ is the epoch drop rate; and $E$ is the total number of training epochs.

## 4.3 AUTO DP-SGD

The automatic DP-SGD algorithm (Auto DP-SGD), which incorporates automatic clipping threshold estimation (AC) and automatic noise multiplier decay, is explained in this section. Auto DP-SGD takes the private dataset, loss function, and other parameters as input. Then, it initializes the model parameters and noise multiplier. The algorithm runs $T$ iterations, where $T = \frac{E}{q}$. $E$ represents the total number of training epochs. $q = B/M$ is the sampling rate, where $B$ is the batch size, and $M$ represents the number of training examples in the data set. In each iteration, the algorithm initializes an empty list $G$. Next, it obtains a batch of data samples that are selected randomly with a sampling rate of $q$. Then, for each sample in the batch, the model gradient is obtained and appended to the list $G$. Auto DP-SGD then calls the Automatic Clipping threshold estimation algorithm (AC) and obtains the average of the clipping threshold and gradients. Now, the noise decay mechanism is called to compute the noise multiplier for the current iteration. The noise decay mechanism $F$ can be one of the four noise decay mechanisms presented in Table 1. The argument $D$ of $F$ is only used for the step noise decay scheduler. The $F$ is set before the algorithm starts. Finally, the Auto DP-SGD updates the average gradient obtained using the AC algorithm and the model parameter. When all iterations have been completed, the algorithm generates the updated model.

We emphasize that the equation to add noise in DP-SGD Abadi et al. (2016): $\tilde{\mathbf{g}}_t = \frac{1}{s}(\sum_i (\bar{\mathbf{g}}_t(x_i) + \mathcal{N}(0, \sigma^2 S^2 \mathbf{I}))$ is the same as our proposed algorithm noise adding mechanism (see the Appendix

A.2 for a more detailed explanation). Noise multiplier decays $F$, scale factor $W$, and learning rate schedulers reduce the magnitude of scaled noisy gradients. The Algorithm 1 (AC) finds the clipping threshold for every sample considering this reduction in the magnitude of the scaled gradients using the total gradient norm. Therefore, Algorithm 2 (Auto DP-SGD) results in a lower sensitivity, as shown in Tables 11-13. Lower sensitivity improves privacy, as discussed in the Appendix A. Lower sensitivity also reduces the noise added to the model, leading to an improvement in accuracy. This is the reason why Auto DP-SGD has obtained better accuracy even with a lower privacy budget ($\epsilon$).

## 5 EXPERIMENTS

In this section, we use the DP and DP-SGD interchangeably. In Tables 2 - 4, the Acc., $\Delta\epsilon$, and $\Delta$Acc. should be considered in percentages. In Tables 2 - 5, Auto DP-S is the Auto DP used with step noise multiplier decay, $\sigma_0$ is the initial noise multiplier for Auto DP-S, dynamic DP, and adaptive DP. For DP, $\sigma_0$ is the noise multiplier that is used throughout the training. We want to emphasize that the lower the privacy budget ($\epsilon$), the higher the privacy and the less vulnerable the model is to inference-based attacks.

***Implementation details.*** The codes for our experiments are developed and implemented using Py-Torch Tor (2017). We conduct all the experiments on a server equipped with an Intel Core i9-10980XE CPU, 251 GB of memory, and four Nvidia Quadro RTX 8000 GPUs, running Ubuntu 18.04 OS. We use the expression given in Section 4.3, Theorem 2 of Zhang et al. (2021) to compute the privacy budget for adaptive DP-SGD Abadi et al. (2016). The adaptive DP-SGD is implemented using the logic shown in Zhang et al. (2021). To implement dynamic DP-SGD, we use the codes provided in dp (2021). We will release the codes for our proposed algorithm after the paper is published or accepted. In this paper, we provide the details to replicate our work.

Table 2: Accuracy (Acc.) and privacy budget for different DP algorithms using MNIST dataset and custom CNN model.

| | DP Abadi et al. (2016) | | Dynamic DP Du et al. (2021) | | ADP Zhang et al. (2021) | | Auto DP-S | | | |
|---|---|---|---|---|---|---|---|---|---|---|
| $\sigma_0$ | $\epsilon(\downarrow)$ | Acc. | $\epsilon(\downarrow)$ | Acc. | $\epsilon(\downarrow)$ | Acc. | $\epsilon(\downarrow)$ | Acc. | $\Delta\epsilon$ | $\Delta$Acc. |
| 1.4929 | 1 | 92.45 | 1 | 94.84 | 9.52 | 93.71 | **0.50** | **99.26** | -50.0 | 4.67 |
| 0.9584 | 2 | 94.01 | 2 | 95.44 | 16.86 | 94.98 | **0.68** | **99.29** | -66.0 | 4.03 |
| 0.6630 | 5 | 95.24 | 5 | 96.15 | 25.44 | 95.81 | **0.52** | **99.31** | -89.6 | 3.29 |
| 0.5517 | 10 | 95.70 | 10 | 96.26 | 32.32 | 96.26 | **0.51** | **99.34** | -94.9 | 3.20 |

Table 3: Accuracy (Acc.) and privacy budget for different DP algorithms using CIFAR10 dataset and pre-trained NFNet-F0 model.

| | DP Abadi et al. (2016) | | Dynamic DP Du et al. (2021) | | ADP Zhang et al. (2021) | | Auto DP-S | | | |
|---|---|---|---|---|---|---|---|---|---|---|
| $\sigma_0$ | $\epsilon(\downarrow)$ | Acc. | $\epsilon(\downarrow)$ | Acc. | $\epsilon(\downarrow)$ | Acc. | $\epsilon(\downarrow)$ | Acc. | $\Delta\epsilon$) | $\Delta$ Acc. |
| 1.6082 | 1 | 92.29 | 1 | 91.31 | 1.87 | 92.58 | **0.50** | **94.96** | -73.26 | 2.57 |
| 1.0134 | 2 | 92.85 | 2 | 92.08 | 3.04 | 93.18 | **0.72** | **95.12** | -76.32 | 2.08 |
| 0.6848 | 5 | 93.29 | 5 | 92.69 | 4.62 | 93.54 | **1.00** | **95.18** | -78.35 | 1.75 |
| 0.5649 | 10 | 93.52 | 10 | 93.07 | 5.71 | 93.77 | **1.19** | **95.24** | -79.16 | 1.57 |

***Computation details.*** Each experiment has taken approximately 20 hours, and this project involves more than 100 successful experiments. The project took five months to test the Auto DP-SGD and to collect all the required experiments.

***Evaluation strategy.*** Specifically, we use a custom CNN model, consisting of two convolutional layers, fully connected layers, a Max Pool layer, and a ReLU, to train on the MNIST data set LeCun et al. (1998). Furthermore, we fine-tune the NFNet-F0 and NFNet-F1 pretrained models Brock et al. (2021) on the CIFAR10 and CIFAR100 datasets Krizhevsky et al. (2009), respectively. In the fine-tuning process, we reinitialized only the final classification layer. We find that existing

Table 4: Accuracy (Acc.) and privacy budget for different DP algorithms using CIFAR100 dataset and pre-trained NFNet-F1 model.

| | DP Abadi et al. (2016) | | Dynamic DP Du et al. (2021) | | ADP Zhang et al. (2021) | | Auto DP-S | | | |
|---|---|---|---|---|---|---|---|---|---|---|
| $\sigma_0$ | $\epsilon(\downarrow)$ | Acc. | $\epsilon(\downarrow)$ | Acc. | $\epsilon(\downarrow)$ | Acc. | $\epsilon(\downarrow)$ | Acc. | $\Delta\,\epsilon$ | $\Delta$ Acc. |
| 1.6082 | 1 | 58.54 | 1 | 65.77 | 3.89 | 70.42 | 1.70 | **79.09** | -56.30 | 12.31 |
| 1.0134 | 2 | 71.20 | 2 | 69.42 | 6.43 | 72.63 | 2.49 | **79.92** | -61.28 | 10.04 |
| 0.6848 | 5 | 73.06 | 5 | 71.37 | 10.03 | 74.43 | 3.47 | **80.30** | -65.40 | 7.89 |
| 0.5649 | 10 | 74.11 | 10 | 72.47 | 12.56 | 75.35 | 4.10 | **80.42** | -67.36 | 6.73 |

Table 5: Accuracy (Acc.) and privacy budget for different DP algorithms using AG News Corpus dataset and BiLSTM model.

| | DP Abadi et al. (2016) | | Adaptive DP Zhang et al. (2021) | | Auto DP-S | | | |
|---|---|---|---|---|---|---|---|---|
| $\sigma_0$ | $\epsilon(\downarrow)$ | Acc. | $\epsilon(\downarrow)$ | Acc. | $\epsilon(\downarrow)$ | Acc. | $\Delta\,\epsilon$ | $\Delta$ Acc. |
| 1.3768 | 1 | 83.63% | 3.78% | 84.57% | 2.04 | **85.18%** | -46.03% | 0.72% |
| 0.9169 | 2 | 84.76% | 5.87% | 85.32% | 3.01 | **86.33%** | -48.72% | 1.18% |
| 0.6585 | 5 | 85.56% | 8.51% | 85.93% | **4.13** | **87.09%** | -51.47% | 1.35% |
| 0.5468 | 10 | 86.09% | 10.53% | 86.13% | **4.91** | **87.35%** | -53.37% | 1.42% |

Table 6: Accuracy of non-private model for different datasets.

| Dataset | Accuracy |
|---|---|
| MNIST | 99.49% |
| CIFAR10 | 95.79% |
| CIFAR100 | 81.21% |
| AG News Corpus | 89.06% |

methods Zhang et al. (2021); Du et al. (2021) are evaluated on simple data sets such as CIFAR10 and MNIST. We make our evaluation more comprehensive by evaluating our Auto DP-SGD on simple and complex datasets, from custom models to pretrained ones on image and text classification tasks. Auto-DP-SGD is independent of the model and dataset, which is the reason why our approach works well on all kinds of models and data sets. Lastly, we conduct experiments using the BiLSTM model on the AG News Corpus dataset AG (2005). The batch size for all algorithms is fixed before the training starts. However, samples are selected in batches at random in every iteration. To make our evaluation robust and fundamentally correct, we use the same set of hyperparameters to run all algorithms on an appropriate dataset.

To perform a comparative analysis, we evaluate the performance of our proposed approach against non-private SGD, DP-SGD Abadi et al. (2016), dynamic DP-SGD Du et al. (2021), and adaptive DP-SGD Zhang et al. (2021). The privacy budget for DP-SGD is calculated using the Renyi DP implementation from TFPrivacy TFP (2019). On the contrary, the loss of privacy of dynamic DP-SGD is calculated using the Gaussian DP method as described in Du et al. (2021). For Auto DP-SGD, the privacy loss is estimated based on the equations provided in Table 8. We conduct experiments with DP-SGD and dynamic DP-SGD for epsilon values of 1, 2, 5, and 10. To determine the epsilon values for adaptive DP-SGD and Auto DP-SGD, we first calculate the noise multiplier using the TF-Privacy package TFP (2019) for epsilon values of 1, 2, 5, and 10. Then, we utilize these noise multipliers in the respective algorithms. We include these noise multipliers in Tables 2 3 4 5. The probability parameter $\delta$ is set to $10^{-5}$ in all experiments. We want to emphasize that because the privacy budget for the Auto DP-SGD depends on the average sensitivity of the algorithm, which becomes known only after training, it is impossible to compute the privacy budget before training.

We find that the suitable clipping threshold for adaptive DP Zhang et al. (2021) and DP Abadi et al. (2016) are 50, 10, 20, and 10 on MNIST, CIFAR10, CIFAR100, and AG News Corpus, respectively, after performing a hyperparameter search. We provide experiments on hyperparameter tuning in Appendix B.2., whereas Appendix B.1. contains details about the datasets used in this work and details of hyperparameters. In Appendix B.3., we analyze the results of the Auto DP-SGD variants, and in Appendix D. we explain how to select the best Auto DP-SGD variant without using validation data to prevent privacy risk.

***Comparative analysis.*** The Adaptive DP results in Table 2 illustrate that a higher clipping threshold (the clipping threshold is set to 50) leads to an increased loss of privacy, with values of $\epsilon = 9.52, 16.86, 25.44, 32.32$ observed. Dynamic DP results in Tables 3 and 4 show poorer performance compared to DP-SGD Abadi et al. (2016). However, it shows better performance in Table 2. With our automatic clipping threshold estimation algorithm (Algorithm 1), the Auto DP approach overcomes the challenge of setting the clipping threshold before training by learning it automatically. Auto DP-S outperforms current state-of-the-art (SOTA) DP-SGD methods on most of the benchmark datasets used in this study. More specifically, Tables 2, 3, 4 and 5 show accuracy improvements of 3.20%, 1.57%, 6.73%, and 1.42% as well as a reduction in the privacy budget of 94.9%, 79.16%, 67.36%, and 53.37% using Auto DP-S. Auto DP-S achieves a higher reduction in privacy budget than other DP methods because it uses a number of techniques to reduce the average sensitivity of the algorithm. These techniques include gradient scaling using automatic clipping threshold estimation algorithm, automatic noise multipliers, and learning rate schedulers. The average sensitivity is directly proportional to the privacy budget ($\epsilon$) (see Appendix A. to understand the reason.). Therefore, by reducing the average sensitivity, Auto DP-S can achieve a much lower privacy budget ($\epsilon$). Auto DP-S achieves accuracy similar to that of the non-private model, even with a lower privacy budget. For example, on MNIST Auto DP-S achieves an accuracy of 99.39% at $\epsilon$ of 0.51, which is only slightly lower than the non-private model's accuracy of 99.49%. The same pattern is repeated for the rest of the datasets considered in the paper. The next closest accuracy to the nonprivate model in MNIST is obtained by dynamic DP, which has an accuracy of 96.26% at $\epsilon$ of 10. Since it is impossible to compute the privacy budget before starting the training for Auto DP-S, we show the improvements in accuracy and privacy in percentages rather than comparing the accuracy for the fixed privacy budget across the algorithms. The important point to note is that the privacy budget formulation for Auto DP-L and adaptive DP Zhang et al. (2021) is identical. However, the privacy budget attributed to Auto DP-L is reduced mainly due to automatic clipping threshold estimation. This algorithm precisely assesses the clipping threshold and decreases the average sensitivity of Auto DP-L. Consequently, this leads to a lower level of privacy leakage.

## 6 LIMITATIONS AND FUTURE WORK

The proposed Auto DP-E incurs a greater privacy loss. For example, on AG News Corpus data, using the BiLSTM model requires a privacy budget ($\epsilon$) 687.61 to obtain 81.06% accuracy for Auto DP-E, while for Auto DP-S it only needs ($\epsilon$) 4.10 to obtain 80.42% accuracy. Adjusting the noise multiplier dynamically according to the characteristics of the data set and the model might be a good direction to improve Auto DP-SGD. We leave differentially private hyperparameter tuning and improving the utility using differentially private neural architecture search (DPNAS) Cheng et al. (2022) to future work. The privacy accountant tCDP can be replaced with an exact computation method, such as numerical methods Gopi et al. (2021), to obtain an exact privacy computation. However, using it is not straightforward when noise and clipping threshold change every epoch.

## 7 CONCLUSION

We demonstrated that the existing adaptive DP-SGD algorithm incurs a higher privacy loss due to a higher clipping threshold and is evaluated on a limited set of benchmarks. Furthermore, the adaptive DP-SGD and dynamic DP-SGD algorithms were not evaluated on the CIFAR-100 dataset in their paper. We then proposed an approach called Auto DP-SGD that automatically estimates the clipping threshold in every iteration using the total gradient norm of the model. Specifically, we proposed the three variants of Auto DP-SGD, namely Auto DP-SGD-T, Auto DP-SGD-S, and Auto DP-SGD-E, and presented Auto DP-SGD-L, which uses our automatic clipping threshold estimation algorithm. Finally, we formalized how to calculate the privacy loss using the tCDP privacy accountant. Auto DP-SGD-S consistently improves the SOTA privacy/utility trade-off on various benchmarks. We then showed that it is possible to improve privacy using learning rate schedulers and a scale factor

with little or no effect on accuracy. Furthermore, we explained how to select the best Auto DP-SGD variant that does not require a privacy budget more than what is needed to train the Auto DP algorithm. The variant of Auto DP-SGD that performs best, Auto DP-SGD-S, improves accuracy by 3.20%, 1.57%, 6.73%, and 1.42% as well as reduces the privacy budget ($\epsilon$) by 94.9%, 79.16%, 67.36%, and 53.37% using MNIST, CIFAR10, CIFAR100, and AG News Corpus, respectively.

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

# A  PROOFS FOR AUTO DP-SGD VARIANTS PRIVACY BUDGET AND NOISE ADDITION

Table 7 shows the notations used in this project.

Table 7: Explanation of different notations used in this work.

| Notation | Explanation |
|---|---|
| $B$ | Batch size |
| $W$ | scale factor |
| $\sigma_0$ | Initial noise multiplier |
| $e$ | Current epoch number |
| $E$ | Total number of training epochs |
| $b$ | Provides sample number in a batch of data samples. |
| $R$ | Decay rate |
| $q$ | Sampling rate |
| $M$ | Total length of the dataset |
| $T$ | Total number of training iterations |
| $t$ | Current iteration number |
| $G$ | Represents a batch of model gradients |
| $\eta_t$ | Learning rate |
| $D$ | Epoch drop rate |
| $N$ | Type of noise multiplier decay mechanism |
| $C$ | Average sensitivity of gradients |
| $\rho, \omega$ | Privacy parameters of tCDP |
| $\rho_e, \omega_e$ | The $\rho, \omega$ at $e^{th}$ epoch. |
| $\rho_{total}, \omega_{total}$ | The total privacy budget parameters under tCDP |
| $\tilde{g}_t$ | Averaged noisy gradient at $t^{th}$ iteration |
| $\bar{g}_t(x_i)$ | Clipped gradient of the $i^{th}$ training data sample at $t^{th}$ iteration |
| $\bar{g}_t$ | The averaged clipped gradient |
| $\mathcal{N}(0, \sigma^2 s^2 \mathbf{I})$ | The gaussian distribution with mean 0 and standard deviation $\sigma s$ |
| $P$ | $P = \frac{E}{D}$ |
| $p$ | An integer that ranges from 0 to $P-1$ |
| $L$ | Total gradient of the norm |
| $AC$ | Automatic Clipping threshold estimation algorithm |

In this work, we mainly use the lemma 1- 4 of tCDP, as demonstrated in Bun et al. (2018) and presented in the background 3 of the article. The lemma 1 establishes a relationship between the Gaussian mechanism and the tCDP privacy accountant. Lemma 2 outlines the composition property of two randomized functions under tCDP. The lemma 3 presents a method for converting the privacy budget in the tCDP accountant to standard $(\epsilon, \delta)$-differential privacy. Finally, the lemma 4 demonstrates privacy amplification through random sampling using tCDP. We utilize these lemmas as the foundation for deriving the proof of our proposed algorithm. Then, using Lemmas 1–4, we prove the privacy parameters for different decay types in Table 8.

Table 8: Total privacy budget for different noise multiplier decay mechanisms.

| Decay type | $\rho_{total}$ | $\omega_{total}$ |
|---|---|---|
| Linear | $\frac{13(B/M)^2 C^2 (1-R^E)}{2\sigma_0^2 (R^{E-1}-R^E)}$ | $\frac{\log(M/B)\sigma_0^2 R^{E-1}}{2C^2}$ |
| Time | $\frac{13(B/M)^2 C^2 (2E+R(E)(E-1))}{4\sigma_0^2}$ | $\frac{\log(M/B)\sigma_0^2}{2C^2(1+R(E-1))}$ |
| Exponential | $\frac{13(B/M)^2 C^2 (e^{RE}-1)}{2\sigma_0^2 (e^R-1)}$ | $\frac{\log(M/B)\sigma_0^2}{2C^2(e^{R(E-1)})}$ |
| Step | $\frac{13(B/M)^2 C^2 (1-R^P)}{2\sigma_0^2 D(R^{P-1}-R^P)}$ | $\frac{\log(M/B)\sigma_0^2 R^{P-1}}{2C^2}$ |

To estimate the cumulative privacy loss of the proposed differentially private stochastic gradient descent (DP-SGD) algorithm, we use the composition theorem of truncated concentrated differential privacy (tCDP), which was created to support more computations and offer a sharper and tighter analysis of privacy loss than the strong composition theorem of $(\epsilon, \delta)$-DP. We provide the cumulative privacy budget for the three types of noise multiplier decay considered in this work: (i) time decay, (ii) exponential decay, and (iii) step decay, and present the total privacy budget for linear decay given by Zhang et al. (2021). Table 8 provides the final expressions to compute the privacy budget for Auto DP-SGD. In Table 8, $\sigma_0$ is the initial noise multiplier; $E$ is the total number of training epochs; $R$ is the decay rate; $M$ is the total number of training samples; $B$ is the batch size; $D$ is the epoch drop rate; $C$ is the average sensitivity of the gradients; and the ratio of the total number of training epochs to the epoch drop rate is $P = E/D$. $C$ is the average clipping threshold that is estimated by averaging the clipping threshold obtained for each sample in each training iteration using algorithm 1.

To derive the final expression for the step decay, we simplified the step decay from $\sigma_e^2 = \sigma_0^2 R^{\lfloor e/D \rfloor}$, $R = 0.5$, $D = 10$ to $\sigma_p^2 = D\sigma_0^2 R^p$ where $p$ ranges from 0 to $P-1$, and we assumed that $E$ is divisible by $D$. After estimating $\rho_{total}$ and $\omega_{total}$ using the lemma 3. Our algorithm satisfies $(\rho_{total} + 2\sqrt{\rho_{total} ln(1/\delta)}, \delta)-$ DP, which means $\epsilon = (\rho_{total} + 2\sqrt{\rho_{total} ln(1/\delta)})$. Table 8 shows that $\rho_{total}$ is directly proportional to the average sensitivity of the gradients ($C$) and $\epsilon$. Therefore, a lower average sensitivity of the gradients improves privacy. After obtaining $\rho_{total}$ and $\omega_{total}$, we can apply the lemma 3 to calculate the corresponding privacy parameters. Specifically, $\epsilon$ should be set to $(\rho_{total} + 2\sqrt{\rho_{total} \ln(1/\delta)})$, where $\delta$ is a predetermined fixed value representing the probability of failure.

### A.1 PROOFS FOR DIFFERENT TYPES OF NOISE MULTIPLIER DECAY MECHANISMS

The notation $\log$ in all subsequent expressions refers to the natural logarithm. The sum of terms in a geometric sequence can be expressed as follows:

$$s_n = \frac{a_1(r^n - 1)}{r - 1}, \quad r > 1 \tag{6}$$

Here, $s_n$ represents the sum of the first $n$ terms of the geometric sequence, $r$ is the common ratio, $a_1$ denotes the first term in the geometric sequence, and $n$ represents the number of terms in the sequence.

The sum of the first $n$ natural numbers is expressed as follows:

$$\Sigma_{i=1}^n i = \frac{n(n+1)}{2} \tag{7}$$

Based on Lemma 1 and Lemma 4, the values of $\rho_e$ and $\omega_e$ are given by:

$$\rho_e = 13(B/M)^2 \left(\frac{C^2}{2\sigma_e^2}\right) \tag{8}$$

$$\omega_e = \frac{log(M/B)\sigma_e^2}{2C^2} \tag{9}$$

Where $(B/M)$ represents the number of entries in a single batch of training examples and $e$ ranges from 0 to $E-1$.

Now, we can compute $\rho_{total}$ and $\omega_{total}$ using Lemma 2 as follows:

$$\rho_{total} = 13(B/M)^2 \left(\frac{C^2}{2}\right) \left(\frac{1}{\sigma_0^2} + \frac{1}{\sigma_1^2} + ... + \frac{1}{\sigma_e^2} + ... + \frac{1}{\sigma_{E-1}^2}\right) \tag{10}$$

$$\omega_{total} = \frac{log(M/B)}{2C^2} min(\sigma_0^2, \sigma_1^2, ..., \sigma_e^2, .., \sigma_{E-1}^2) \tag{11}$$

### A.1.1 PROOF FOR LINEAR NOISE MULTIPLIER DECAY MECHANISM

According to the linear noise multiplier decay mechanism Zhang et al. (2021):

$$\sigma_e^2 = R\sigma_{e-1}^2, R \in (0,1) \tag{12}$$

Now, let us substitute Equation 12 into Equation 10:

$$\rho_{total} = 13(B/M)^2(\frac{C^2}{2})(\frac{1}{\sigma_0^2} + \frac{1}{R\sigma_0^2} + \frac{1}{R^2\sigma_0^2} + ... + \frac{1}{R^e\sigma_0^2} + ... + \frac{1}{R^{E-1}\sigma_0^2}) \tag{13}$$

$$\rho_{total} = 13(B/M)^2(\frac{C^2}{2\sigma_0^2})(1 + \frac{1}{R} + \frac{1}{R^2} + ... + \frac{1}{R^e} + ... + \frac{1}{R^{E-1}}) \tag{14}$$

Equation 14 can be summarized as follows:

$$\rho_{total} = 13(B/M)^2(\frac{C^2}{2\sigma_0^2})(\Sigma_{e=0}^{E-1}\frac{1}{R^e}) \tag{15}$$

Using the sum of the terms of the geometric sequence formula 6, the equation 15 can be summarized as follows:

$$\rho_{total} = 13(B/M)^2(\frac{C^2}{2\sigma_0^2})(\frac{1.(\frac{1}{R})^E - 1}{\frac{1}{R} - 1}) \tag{16}$$

After simplifying the equation 16 further:

$$\rho_{total} = 13(B/M)^2(\frac{C^2}{2\sigma_0^2})(\frac{1 - R^E}{R^{E-1} - R^E}) \tag{17}$$

Now, substitute the linear noise multiplier decay 12 into the equation 11

$$\omega_{total} = \frac{log(M/B)}{2C^2}min(\sigma_0^2, R\sigma_0^2, ..., R^e\sigma_0^2, ..., R^{E-1}\sigma_0^2) \tag{18}$$

$$\omega_{total} = \frac{log(M/B)\sigma_0^2}{2C^2}min(1, R, ..., R^e, ..., R^{E-1}) \tag{19}$$

Since $R \in (0,1)$, equation 19 becomes as follows:

$$\omega_{total} = \frac{log(M/B)\sigma_0^2}{2C^2}(R^{E-1}) \tag{20}$$

### A.1.2 PROOF FOR TIME NOISE MULTIPLIER DECAY MECHANISM

The time noise multiplier decay mechanism is expressed as follows:

$$\sigma_e^2 = \frac{\sigma_0^2}{1 + Re}, R = \frac{\sigma_0}{T}, R \in (0,1) \tag{21}$$

Substituting equation 21 in equation 10. Equation 10 looks like follows:

$$\rho_{total} = 13(B/M)^2\frac{C^2}{2}(\frac{1}{\sigma_0^2} + \frac{1+R}{\sigma_0^2} + \frac{1+2R}{\sigma_0^2} + ... + \frac{1+Re}{\sigma_0^2} + ... + \frac{1+R(E-1)}{\sigma_0^2}) \tag{22}$$

$$\rho_{total} = 13(B/M)^2(\frac{C^2}{2\sigma_0^2})[1 + (1+R) + (1+2R) + ... + (1+eR) + ... + (1+(E-1)R)] \quad (23)$$

The equation 23 can be summarized as follows:

$$\rho_{total} = 13(B/M)^2(\frac{C^2}{2\sigma_0^2})[\Sigma_{e=0}^{E-1}(1+Re)] \quad (24)$$

In simplifying, equation 24 using equation 7 becomes as follows:

$$\rho_{total} = 13(B/M)^2(\frac{C^2}{2\sigma_0^2})[E + \frac{R(E)(E-1)}{2}] \quad (25)$$

$$\rho_{total} = 13(B/M)^2(\frac{C^2}{4\sigma_0^2})[2E + R(E)(E-1)] \quad (26)$$

Now, substitute the decay of the time noise multiplier 21 into the equation 11

$$\omega_{total} = \frac{log(M/B)}{2C^2} min(\sigma_0^2, \frac{\sigma_0^2}{1+R}, ...., \frac{\sigma_0^2}{1+Re}, ...., \frac{\sigma_0^2}{1+R(E-1)}) \quad (27)$$

$$\omega_{total} = \frac{log(M/B)\sigma_0^2}{2C^2} min(1, \frac{1}{1+R}, ...., \frac{1}{1+Re}, ...., \frac{1}{1+R(E-1)}) \quad (28)$$

In further simplification, Equation 28 becomes as:

$$\omega_{total} = \frac{log(M/B)\sigma_0^2}{2C^2(1+R(E-1))} \quad (29)$$

### A.1.3 PROOF FOR STEP NOISE MULTIPLIER DECAY MECHANISM

To derive the final expression for the step decay, we simplify the step decay from Equation 30 to Equation 36 under the assumption that $E$ is divisible by $D$ and $P = E/D$.

$$\sigma_e^2 = \sigma_0^2 R^{\lfloor e/D \rfloor}, R \in (0,1) \quad (30)$$

To illustrate the transformation of Equation 30, let us consider $E = 100$, $D = 10$, and $P = \frac{E}{D} = \frac{100}{10} = 10$. Now, using Equation 30, we can express $\sigma_e^2$ for $e$ that varies from 0 to $E - 1 = 99$ as follows:

$$\sigma_0^2 = \sigma_0^2 R^{\lfloor 0/10 \rfloor} = \sigma_0^2, \sigma_1^2 = \sigma_0^2 R^{\lfloor 1/10 \rfloor} = \sigma_0^2, ..., \sigma_{10}^2 = \sigma_0^2 R^{\lfloor 10/10 \rfloor} = \sigma_0^2 R,$$
$$\sigma_{11}^2 = \sigma_0^2 R^{\lfloor 11/10 \rfloor} = \sigma_0^2 R, ..., \sigma_{98}^2 = \sigma_0^2 R^{\lfloor 98/10 \rfloor} = \sigma_0^2 R^9, \sigma_{99}^2 = \sigma_0^2 R^{\lfloor 99/10 \rfloor} = \sigma_0^2 R^9 \quad (31)$$

Now, the sum of the noise multiplier at all the epochs is equal to:

$$\Sigma_{e=0}^{99}\sigma_e^2 = 10\sigma_0^2 + 10R\sigma_0^2 + ... + 10R^9\sigma_0^2 \quad (32)$$

Equation 32 can be generalized as follows:

$$\Sigma_{e=0}^{E-1}\sigma_e^2 = D\sigma_0^2 + DR\sigma_0^2 + ... + DR^p\sigma_0^2 + ... + DR^{P-1}\sigma_0^2 \quad (33)$$

$$\Sigma_{e=0}^{E-1}\sigma_e^2 = \Sigma_{p=0}^{P-1}DR^p\sigma_0^2 \quad (34)$$

To simplify the formula, let us define the following:

$$\Sigma_{e=0}^{E-1}\sigma_e^2 = \Sigma_{p=0}^{P-1}\sigma_p^2 \tag{35}$$

Then, $\sigma_p^2$ can be expressed as follows:

$$\sigma_p^2 = D\sigma_0^2 R^p \tag{36}$$

Equation 10 can be generalized as follows:

$$\rho_{total} = 13(B/M)^2(\frac{C^2}{2})(\Sigma_{e=0}^{E-1}\frac{1}{\sigma_e^2}) \tag{37}$$

Using Equation 35, Equation 37 can be expressed as follows:

$$\rho_{total} = 13(B/M)^2(\frac{C^2}{2})(\Sigma_{p=0}^{P-1}\frac{1}{\sigma_p^2}) \tag{38}$$

Now, let us substitute Equation 36 into Equation 38:

$$\rho_{total} = 13(B/M)^2(\frac{C^2}{2D\sigma_0^2})(\Sigma_{p=0}^{P-1}\frac{1}{R^p}) \tag{39}$$

In expanding Equation 39, it becomes as follows:

$$\rho_{total} = 13(B/M)^2(\frac{C^2}{2D\sigma_0^2})(1 + \frac{1}{R} + \frac{1}{R^2} + ... + \frac{1}{R^p} + ... + \frac{1}{R^{P-1}}) \tag{40}$$

Using the formula for the sum of terms in a geometric sequence, Equation 6, we can summarize Equation 40 as follows:

$$\rho_{total} = 13(B/M)^2(\frac{C^2}{2D\sigma_0^2})(\frac{1.(\frac{1}{R})^P - 1}{\frac{1}{R} - 1}) \tag{41}$$

After further simplifying Equation 41:

$$\rho_{total} = 13(B/M)^2(\frac{C^2}{2D\sigma_0^2})(\frac{1 - R^P}{R^{P-1} - R^P}) \tag{42}$$

Now, let us substitute the revised step noise multiplier decay (Equation 36) into Equation 42:

$$\omega_{total} = \frac{log(M/B)}{2C^2}min(D\sigma_0^2, RD\sigma_0^2, ..., R^p D\sigma_0^2, ..., R^{P-1}D\sigma_0^2) \tag{43}$$

$$\omega_{total} = \frac{log(M/B)D\sigma_0^2}{2C^2}min(1, R, ..., R^p, ..., R^{P-1}) \tag{44}$$

Since $R < 1$, we have:

$$\omega_{total} = \frac{log(M/B)D\sigma_0^2}{2C^2}(R^{P-1}) \tag{45}$$

A.1.4 PROOF FOR EXPONENTIAL NOISE MULTIPLIER DECAY MECHANISM

The exponential noise multiplier decay mechanism is expressed as follows:

$$\sigma_t^2 = \sigma_0^2 \times e^{-Re}, \quad R \in (0, 1) \tag{46}$$

Now, let us substitute Equation 46 into Equation 10:

$$\rho_{total} = 13(B/M)^2 (\frac{C^2}{2})(\frac{1}{\sigma_0^2} + \frac{1}{e^{-R}\sigma_0^2} + ... + \frac{1}{e^{-Re}\sigma_0^2} + ... + \frac{1}{e^{-(E-1)R}\sigma_0^2}) \tag{47}$$

$$\rho_{total} = 13(B/M)^2 (\frac{C^2}{2\sigma_0^2})(1 + e^R + ... + e^{Re} + ... + e^{(E-1)R}) \tag{48}$$

Equation 48 can be summarized as follows:

$$\rho_{total} = 13(B/M)^2 (\frac{C^2}{2\sigma_0^2})(\Sigma_{e=0}^{E-1} e^{Re}) \tag{49}$$

Using the geometric sequence formula (Equation 14), Equation 49 becomes as follows, considering that $e^R > 1$:

$$\rho_{total} = 13(B/M)^2 (\frac{C^2}{2\sigma_0^2})(\frac{e^{RE} - 1}{e^R - 1}) \tag{50}$$

After substituting Equation 46 into Equation 11, it can be expressed as follows:

$$\omega_{total} = \frac{log(M/B)}{2C^2} min(\sigma_0^2, \sigma_0^2 e^{-R}, ..., \sigma_0^2 e^{-eR}, ..., \sigma_0^2 e^{-(E-1)R}) \tag{51}$$

$$\omega_{total} = \frac{log(M/B)\sigma_0^2}{2C^2} min(1, \frac{1}{e^R}, ..., \frac{1}{e^{eR}}, ..., \frac{1}{e^{(E-1)R}}) \tag{52}$$

Equation 52 can be simplified considering the fact that $\frac{1}{e^R} < \frac{1}{e^{eR}} < \frac{1}{e^{(E-1)R}}$ :

$$\omega_{total} = \frac{log(M/B)\sigma_0^2}{2C^2 e^{(E-1)R}} \tag{53}$$

A.2 EQUIVALENCE OF NOISE ADDITION MECHANISM TO DP-SGD

This section aims to prove that the noise addition mechanism of the proposed algorithm is equivalent to the noise addition mechanism used in DP-SGD Abadi et al. (2016).

The noise addition mechanism in DP-SGD is given by:

$$\tilde{\mathbf{g}}_t = \frac{1}{B}(\sum_i (\bar{\mathbf{g}}_t(x_i) + \mathcal{N}(0, \sigma_t^2 s^2 \mathbf{I})) \tag{54}$$

where $\sigma_t$ is the same for every iteration in the DP-SGD algorithm.

The noise addition mechanism of the Auto DP-SGD algorithm is given by:

$$\tilde{\mathbf{g}}_t = \bar{g}_t + \frac{\mathcal{N}(0, \sigma_t^2 C^2 \mathbf{I})}{B} \tag{55}$$

The expanded form of Equation 54, considering different clipping thresholds for every sample in a batch of gradients, is given by:

$$\tilde{\mathbf{g}}_t = \frac{1}{B}\sum_i (\bar{\mathbf{g}}_t(x_i)) + \frac{1}{B}(\mathcal{N}(0, \sigma_t^2 \mathbf{I})(\frac{s_0 + s_1 + ...s_b + ... + s_{B-1}}{B})) \tag{56}$$

Where $s_0, s_1, ..., s_b, ..., s_B$ are the sensitivities (clipping threshold) of the $0, 1, ..., b, ..., B^{th}$ data sample in the batch.

Let $C = \frac{s_0 + s_1 + ... + s_{B-1}}{B}$. Then, Equation 56 can be written as follows, which is the proposed algorithm noise-adding mechanism:

$$\tilde{\mathbf{g}}_t = \frac{1}{B} \sum_i (\bar{\mathbf{g}}_t(x_i)) + \frac{1}{B}(\mathcal{N}(0, \sigma_t^2 C^2 \mathbf{I})) \tag{57}$$

The proposed Auto DP-SGD algorithm differs from DP-SGD Abadi et al. (2016) in two aspects. First, it utilizes automatic noise multipliers, which adjust the noise multiplier for each iteration and automatically determine the clipping threshold for each sample. This requires averaging the clips over the batch of data samples to compute the average sensitivity. In contrast, DP-SGD uses a fixed clipping threshold that is the same for all data samples in the batch. Therefore, the clipping threshold applied to a single data sample is equivalent to the average clipping threshold. Second, Auto DP-SGD automates the noise multiplier throughout the training process, whereas DP-SGD uses a constant noise multiplier throughout the training.

## B  EXPERIMENTS ON HYPER-PARAMETERS AND AUTO DP-SGD VARIANTS

### B.1  DATASET DETAILS

This section provides details on the data sets used in this study.

**MNIST.** The MNIST dataset LeCun et al. (1998) consists of grayscale images with dimensions of $28 \times 28$ pixels. The training set comprises 60,000 images, while the test set contains 10,000 images.

**CIFAR10.** The CIFAR-10 dataset Krizhevsky et al. (2009) consists of 60,000 color images with dimensions of $32 \times 32$ pixels. It includes 6,000 images per class, spanning across 10 classes. The data set is divided into 50,000 training images and 10,000 test images.

**CIFAR100.** The CIFAR-100 data set Krizhevsky et al. (2009) consists of 60,000 images divided into 100 classes, with 600 images per class. Like CIFAR-10, the CIFAR-100 data set also includes 50,000 training images and 10,000 test images.

During experiments, we ran the above three datasets for 100 training epochs and set the batch size to 64.

**AG News Corpus.** The AG News Corpus data set is a classification data set created by selecting the four most significant classes from the original AG corpus. Each class consists of 30,000 training samples and 1,900 testing samples. In total, there are 120,000 training samples and 7,600 testing samples available. The AG News Corpus dataset is trained for 40 epochs with a batch size of 256.

For Tables 9 and 10, we use the SGD optimizer with a noise multiplier of 2.8 and a learning rate of $10^{-4}$. To generate the results in Table 11, we use an AdamW optimizer with a weight decay of $10^{-3}$, a scale factor of 1.0, a noise multiplier of 2.8, and an initial learning rate of $10^{-3}$. In the case of Table 12, we use an AdamW optimizer with a weight decay of $10^{-3}$, and a noise multiplier of 2.8, and implement a one-cycle learning rate policy with a learning rate set at $10^{-3}$. Additionally, for all other tables in the Appendix and the main paper, we use a scale factor value of 1.0, an AdamW optimizer with a weight decay of $10^{-3}$, and implement a one-cycle learning rate policy with a learning rate set at $10^{-3}$.

### B.2  EXPERIMENTS ON HYPER-PARAMETER TUNING

#### B.2.1  EFFECT OF CLIPPING THRESHOLD ON DP-SGD ABADI ET AL. (2016) AND ADAPTIVE DP-SGD ALGORITHMS ZHANG ET AL. (2021)

Tables 9 and 10 illustrate the impact of the clipping threshold on the DP-SGD algorithm Abadi et al. (2016) and adaptive DP-SGD Zhang et al. (2021) respectively. The results demonstrate that choosing an appropriate clipping threshold involves extensive hyperparameter tuning. Even with hyperparameter adaptation, it is challenging to estimate the optimal clipping threshold. The best clipping threshold for the MNIST, CIFAR10 and CIFAR100 datasets was found to be 50, 10, and

Table 9: Impact of clipping threshod DP-SGD algorithm Abadi et al. (2016).

| Clipping Threshold | MNIST Accuracy | CIFAR10 Accuracy | CIFAR100 Accuracy |
| --- | --- | --- | --- |
| 0.1 | 12.23% | 30.38% | 1.24% |
| 0.25 | 25.86% | 63.89% | 1.96% |
| 0.5 | 52.49% | 80.42% | 4.18% |
| 0.75 | 56.79% | 84.47% | 7.46% |
| 1.0 | 59.07% | 85.88% | 11.46% |
| 2.5 | 72.08% | 88.58% | 35.73% |
| 5 | 84.29% | 89.98% | 50.80% |
| 7.5 | 87.48% | 90.81% | 55.53% |
| 10.0 | 88.68% | **91.09%** | 57.91% |
| 20.0 | - | - | **61.88%** |
| 50.0 | **91.87%** | 89.70% | 46.00% |
| 100.0 | 88.44% | 84.97% | 23.31% |

Table 10: Impact of the clipping threshold on adaptive DP-SGD algorithm Zhang et al. (2021).

| Clipping Threshold | MNIST Accuracy | CIFAR10 Accuracy | CIFAR100 Accuracy |
| --- | --- | --- | --- |
| 0.1 | 12.23% | 30.44% | 1.25% |
| 0.25 | 25.86% | 63.88% | 1.97% |
| 0.5 | 52.49% | 80.48% | 4.12% |
| 0.75 | 56.79% | 84.43% | 7.45% |
| 1.0 | 59.07% | 86.03% | 11.51% |
| 2.5 | 72.08% | 88.63% | 36.29% |
| 5 | 84.29% | 89.98% | 51.67% |
| 7.5 | 87.48% | 90.86% | 56.88% |
| 10.0 | 88.68% | **91.25%** | 59.39% |
| 20.0 | - | - | **63.74%** |
| 50.0 | **91.87%** | 90.81% | 52.52% |
| 100.0 | 88.44% | 87.43% | 31.98% |

20, respectively. When the clipping threshold is set too low, the accuracy decreases significantly due to information loss. On the other hand, setting the clipping threshold too high leads to increased noise, leading to reduced accuracy. Therefore, it is crucial to design algorithms to approximate the optimal clipping threshold and maintain privacy efficiently.

### B.2.2 EFFECT OF LEARNING RATE (LR) SCHEDULERS ON THE AUTO DP-SGD-L ALGORITHM

Learning rate schedulers (LR schedulers) are commonly employed in non-private DL techniques to enhance model accuracy and accelerate convergence. The learning rate determines the step size towards the global minimum. Using a fixed learning rate is not recommended, as gradients often become smaller through training. Learning rate schedulers can improve the performance of deep learning models Bengio (2012) by dynamically changing the learning rate through training. This study investigates the impact of 11 different learning rate schedulers (StepLR, MultiStepLR, ConstantLR, LinearLR, ExponentialLR, PolynomialLR, CosineAnnealingWarmRestarts, CyclicLR, OneCycleLR and ReduceLROnPlateau) on the performance of Auto DP-SGD-L using the MNIST dataset. It has been empirically demonstrated that integrating AdamW with learning rate schedulers yields better results in non-private settings Loshchilov & Hutter (2017). The performance of LR schedulers in nonprivate settings, particularly with AdamW, served as an inspiration for investigating their effects in DP settings. Table 11 shows that all LR schedulers improve sensitivity and accuracy, the OCL LR scheduler being the most effective, achieving an accuracy of 98.02% and sensitivity of 12.2752. As discussed in the Appendix A., lower sensitivity values indicate better privacy. Therefore, LR schedulers enhance the accuracy and privacy of the Auto DP-SGD-L algorithm.

Table 11: Impact of LR schedulers on the Auto DP-SGD-L algorithm.

| LR scheduler | Accuracy | Sensitivity |
|---|---|---|
| StepLR(st) | 97.59% | 13.2799 |
| MultiStepLR(mst) | 97.70% | 13.9652 |
| ConstantLR (con) | 96.49% | 17.7517 |
| LinearLR(li) | 96.23% | 19.0044 |
| ExponentialLR(exp) | 92.43% | 20.6112 |
| PolynomialLR(poly) | 96.86% | 15.1552 |
| CosineAnnealingLR(cos) | 97.92% | 13.0260 |
| CosineAnnealingWarmRestarts(coswr) | 97.43% | 15.8700 |
| CyclicLR(cyc) | 97.61% | 13.1589 |
| OneCycleLR(ocl) | **98.02%** | **12.2752** |
| ReduceLRPlateau(rop) | 97.85% | 14.7475 |
| $LR(10^{-2})$ | 85.63% | 289.5400 |
| $LR(10^{-3})$ | 95.75% | 16.7381 |
| $LR(10^{-4})$ | 96.60% | 24.6880 |

### B.2.3 EFFECT OF SCALE FACTOR ON THE AUTO DP-SGD-L ALGORITHM

To investigate the impact of the scale factor on the Auto DP-SGD-L algorithm, we experimented with five different scale factors: 1.0, 0.9, 0.75, 0.5, and 0.2. Table 12 results show that as the scale factor decreases, the sensitivity improves for all data sets (MNIST, CIFAR10, CIFAR100, and AG News corpus). The lowest sensitivity is 2.4475, 0.9020, 3.0859, and 1.4779 for the respective data sets. Interestingly, not much variation in accuracy is observed across the different scale factors. Based on these findings, it is recommended to use a lower scale factor to enhance privacy without sacrificing accuracy.

Table 12: Impact of scale factor on the Auto DP-SGD-L algorithm.

| Dataset | scale factor | Accuracy | sensitivity |
|---|---|---|---|
| | 1.0 | 98.02% | 12.2752 |
| | 0.9 | 98.06% | 11.0005 |
| MNIST | 0.75 | 98.08% | 9.1679 |
| | 0.5 | 98.01% | 6.1189 |
| | 0.2 | **98.09%** | **2.4475** |
| | 1.0 | 93.68% | 4.5261 |
| | 0.9 | 93.68% | 4.0735 |
| CIFAR10 | 0.75 | 93.68% | 3.3946 |
| | 0.5 | **93.68%** | 2.2630 |
| | 0.2 | 93.64% | **0.9020** |
| | 1.0 | 67.34% | 15.3871 |
| | 0.9 | 67.34% | 13.8484 |
| CIFAR100 | 0.75 | 67.34% | 11.5403 |
| | 0.5 | 67.34% | 7.6935 |
| | 0.2 | **67.70%** | **3.0859** |
| | 1.0 | 82.63% | 7.3702 |
| | 0.9 | 81.89% | 6.7131 |
| AG NEWS CORPUS | 0.75 | 82.36% | 5.6046 |
| | 0.5 | **82.70%** | 3.6476 |
| | 0.2 | 82.09% | **1.4779** |

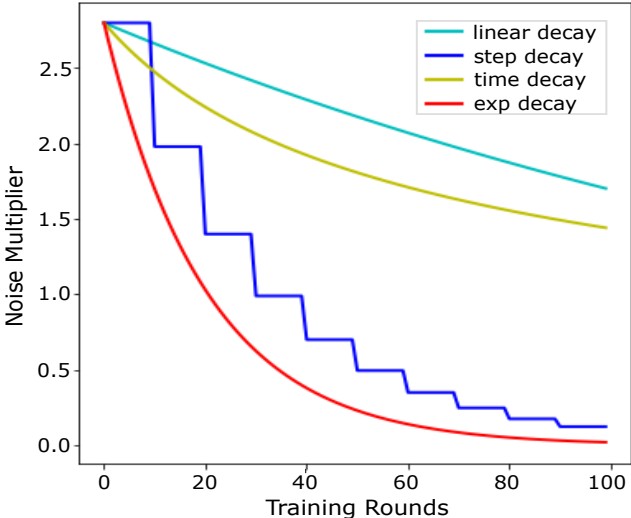

Figure 1: Types of noise multiplier decay mechanisms

Table 13: Impact of noise multiplier decay on the Auto DP-SGD algorithm.

| Dataset | Noise multiplier decay | Accuracy | sensitivity |
|---------|------------------------|----------|-------------|
| MNIST | Linear decay Zhang et al. (2021) | 98.02% | 12.2752 |
| | Exponential decay | **99.36%** | **4.7777** |
| | Time decay | 98.23% | 10.6461 |
| | Step decay | 99.11% | 6.6427 |
| CIFAR10 | Linear decay Zhang et al. (2021) | 93.68% | 4.5261 |
| | Exponential decay | **95.18%** | **3.7083** |
| | Time decay | 93.83% | 4.4091 |
| | Step decay | 94.91% | 4.0232 |
| CIFAR100 | Linear decay Zhang et al. (2021) | 67.34% | 15.3870 |
| | Exponential decay | **80.29%** | **11.6984** |
| | Time decay | 68.84% | 14.9671 |
| | Step decay | 77.84% | 13.1365 |
| AG NEWS CORPUS | Linear decay Zhang et al. (2021) | 82.48% | 7.3702 |
| | Exponential decay | **85.22%** | **7.1516** |
| | Time decay | 83.66% | 7.3594 |
| | Step decay | 84.21% | 7.3065 |

### B.2.4 EFFECT OF NOISE MULTIPLIER DECAY MECHANISMS ON THE AUTO DP-SGD ALGORITHM

Figure 1 illustrates the progression of noise multiplier decay processes throughout the training period. Among the decay mechanisms investigated (linear, time, step, and exponential decay), exponential decay exhibits the least sensitivity and the highest accuracy, followed by step, time, and linear decay in that order. Table 13 summarizes the sensitivity and accuracy of each decay mechanism. In particular, higher decay rates correspond to lower sensitivity and higher accuracy. For example, in the case of the MNIST dataset, the sensitivity for linear, time, step, and exponential decay is 12.2752, 10.6427, 6.6427, and 4.7772, respectively, while the accuracy is 98.02%, 98.23%, 99.11%, and 99.36%, respectively.

Table 14: Accuracy and privacy budget of different Auto DP-SGD variants using MNIST dataset and custom CNN model.

|  | Auto DP-L | | Auto DP-T | | Auto DP-S | | Auto DP-E | |
| --- | --- | --- | --- | --- | --- | --- | --- | --- |
| Noise Multiplier | $\epsilon(\downarrow)$ | Accuracy | $\epsilon(\downarrow)$ | Accuracy | $\epsilon(\downarrow)$ | Accuracy | $\epsilon(\downarrow)$ | Accuracy |
| 1.4929 | 1.00 | 98.62 | 0.98 | 98.74 | 0.50 | 99.26 | 22.85 | 99.30 |
| 0.9584 | 1.49 | 99.03 | 0.84 | 99.08 | 0.68 | 99.29 | 34.20 | 99.33 |
| 0.6630 | 0.85 | 99.16 | 0.72 | 99.17 | 0.52 | 99.31 | 26.65 | 99.35 |
| 0.5517 | 0.81 | 99.23 | 0.67 | 99.25 | **0.51** | 99.34 | 27.18 | **99.36** |

Table 15: Accuracy and privacy budget of different Auto DP-SGD variants using CIFAR10 dataset and pre-trained NFNet-F0 model.

|  | Auto DP-L | | Auto DP-T | | Auto DP-S | | Auto DP-E | |
| --- | --- | --- | --- | --- | --- | --- | --- | --- |
| Noise Multiplier | $\epsilon(\downarrow)$ | Accuracy | $\epsilon(\downarrow)$ | Accuracy | $\epsilon(\downarrow)$ | Accuracy | $\epsilon(\downarrow)$ | Accuracy |
| 1.6082 | 0.72 | 94.22 | 0.73 | 94.27 | 0.50 | 94.96 | 30.25 | 95.13 |
| 1.0134 | 1.03 | 94.45 | 0.96 | 94.58 | 0.72 | 95.12 | 50.65 | 95.18 |
| 0.6848 | 1.44 | 94.55 | 1.28 | 94.65 | 1.00 | 95.18 | 81.79 | 95.21 |
| 0.5649 | 1.72 | 94.74 | 1.43 | 94.82 | **1.19** | 95.24 | 105.06 | **95.29** |

Table 16: Accuracy and privacy budget of different Auto DP-SGD variants using CIFAR100 dataset and pre-trained NFNet-F1 model.

|  | Auto DP-L | | Auto DP-T | | Auto DP-S | | Auto DP-E | |
| --- | --- | --- | --- | --- | --- | --- | --- | --- |
| Noise Multiplier | $\epsilon(\downarrow)$ | Accuracy | $\epsilon(\downarrow)$ | Accuracy | $\epsilon(\downarrow)$ | Accuracy | $\epsilon(\downarrow)$ | Accuracy |
| 1.6082 | 2.57 | 72.24 | 2.62 | 72.4 | 1.70 | 79.09 | 95.94 | 80.6 |
| 1.0134 | 3.77 | 73.95 | 3.50 | 74.37 | 2.49 | 79.92 | 300.35 | 80.78 |
| 0.6848 | 5.29 | 75.59 | 4.57 | 76.14 | 3.47 | 80.3 | 520.24 | 80.98 |
| 0.5649 | 6.26 | 76.26 | 5.26 | 76.87 | **4.10** | 80.42 | 687.61 | **81.06** |

Table 17: Accuracy and privacy budget of different Auto DP-SGD variants using AG News Corpus dataset and BiLSTM model.

|  | Auto DP-L | | Auto DP-T | | Auto DP-S | | Auto DP-E | |
| --- | --- | --- | --- | --- | --- | --- | --- | --- |
| Noise Multiplier | $\epsilon(\downarrow)$ | Accuracy | $\epsilon(\downarrow)$ | Accuracy | $\epsilon(\downarrow)$ | Accuracy | $\epsilon(\downarrow)$ | Accuracy |
| 1.3768 | 2.73 | 84.86 | 2.69 | 85.17 | 2.04 | 85.18 | 247.63 | 87.01 |
| 0.9169 | 4.13 | 86.09 | 3.64 | 86.93 | 3.01 | 86.33 | 457.32 | 87.07 |
| 0.6585 | 5.61 | 86.32 | 4.86 | 86.5 | 4.13 | 87.09 | 800.33 | 87.76 |
| 0.5468 | 6.73 | 86.70 | 5.76 | 86.88 | **4.91** | 87.35 | 1087.63 | **87.78** |

## B.3 EXPERIMENT ANALYSIS ON AUTO DP-SGD VARIANTS

This section compares the results of all the Auto DP variants: Auto DP-L, Auto DP-T, Auto DP-S, and Auto DP-E, where L, T, S, and E represent linear, time, step, and exponential decay mechanisms, respectively. Tables 14 - 17 clearly show that Auto DP-E incurs a significant loss of privacy. For example, the BiLSTM model spent $\epsilon = 1087.63$ to achieve an accuracy of 87.78% on AG News corpus data, as shown in Table 17. Similarly, the NFNet-F1 model used $\epsilon = 687.61$ to reach an accuracy of 81. 06% in CIFAR100, as shown in Table 16. Likewise, other tables 14 15 show the highest privacy leaks in Auto DP-E. The substantial privacy loss of Auto DP-E could be attributed to the exponential decrease in the noise multiplier over iterations, resulting in inadequate noise addition. Among the Auto DP-SGD-L, Auto DP-SGD-T, and Auto DP-SGD-S variants, Auto DP-SGD-S demonstrates the highest performance, followed by Auto DP-SGD-T and Auto DP-SGD-L,

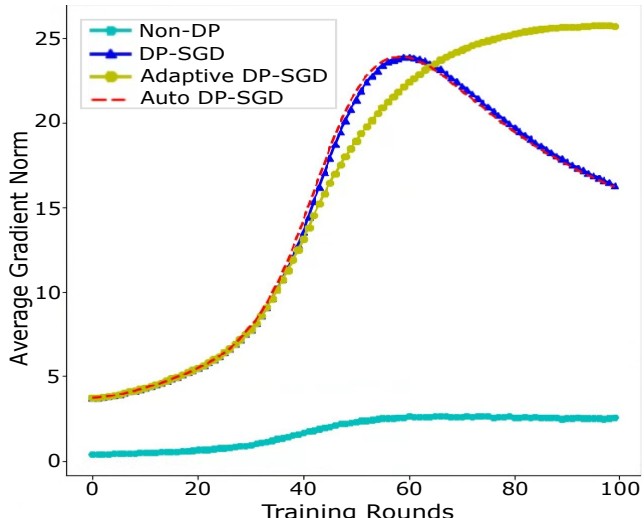

Figure 2: Avg Gradient Norm of different DP algorithms and SGD without DP protection

respectively. For example, consider Table 15, which illustrates that Auto DP-S achieves an accuracy of 95.24% at $\epsilon = 1.19$, while Auto DP-T and Auto DP-L achieve accuracies of 94.82% and 94.74% at $\epsilon = 1.43$ and $\epsilon = 1.72$, respectively. This pattern is consistent across all other datasets, as observed in the respective tables.

## C  IMPORTANCE OF AUTOMATIC CLIPPING THRESHOLD ESTIMATION ALGORITHM

Figure 2 illustrates the progress of the average gradient norm for different DP algorithms and the non-private algorithm during training. Adaptive DP-SGD Zhang et al. (2021) is increasing starting at $20^{th}$ epoch. From Figure 2, it is evident that DP-SGD Abadi et al. (2016) and the Auto DP-SGD algorithm started to grow from a $30^{th}$ epoch and then tends to decrease after a $70^{th}$ epoch. Therefore, the takeaway is that the average gradient norm changes during training and that the clipping threshold should adapt to this change to produce an effective DP model. DP-SGD and adaptive DP-SGD use the constant clipping threshold without considering this gradient behavior. Using the constant clipping threshold, gradients are clipped only in some training iterations where the magnitude of the gradient norm is higher than the clipping threshold. Moreover, a fixed clipping threshold can lead to a more noisy model when the gradients become much smaller in most of the training iterations. This can suppress the gradients and make the model give useless predictions. Our proposed approach considers the average gradient norm, chooses the clipping threshold automatically, and adds noise to the model efficiently. Furthermore, selecting an applicable clipping threshold requires a lot of tuning and leads to a greater loss of privacy, according to Koskela & Honkela (2020). The proposed automatic clipping threshold estimation Algorithm 1 avoids tuning the clipping threshold. We use the MNIST LeCun (1998) dataset and a custom four-layer CNN model to compute the average grade norm.

## D  HOW TO SELECT THE NOISE MULTIPLIER SCHEDULER WITHOUT PRIVACY LEAKAGE?

In Appendix B.2.4., we discussed that a higher accuracy could be obtained using the noise decay scheduler, which has a higher decay rate (we show empirically that a higher decay rate yields lower sensitivity and higher accuracy). A decay scheduler with a higher decay rate can be easily found by plotting the noise multiplier vs. training rounds. In the Appendix, Fig. 1 shows that exponential decay has a higher decay rate, followed by step, time, and linear, respectively. To plot Fig. 1, we use the expressions given in Table 1. Next, consider the results of the table 17 last row that are obtained

using the same set of hyperparameters. Auto DP-E, Auto DP-S, Auto DP-T, and Auto DP-L obtain an accuracy of 86.70%, 86.88%, 87.35%, and 87.88%. Therefore, it is clear that the noise decay scheduler with a higher decay rate gives greater accuracy. Using the expressions of Table 8, we calculate $\epsilon$ using the batch size of 64, the initial noise multiplier, and a clip of 1, and the number of training epochs is set to 100. The $\epsilon$ we get for linear, time, step, and exponential decay is 0.24, 0.23, 0.19, and 9.99, respectively. The exponential decay has a huge privacy leakage compared to other decay methods. Since a lower $\epsilon$ means better privacy, the step decay yields better privacy. Therefore, the step-decay noise scheduler provides the best trade-off between privacy and accuracy among the four noise decay schedulers. All experiments in the main paper validate and demonstrate that the Auto DP-S has a better privacy and accuracy trade-off. Therefore, to choose the better Auto DP-SGD variant, it is unnecessary to use the validation data and can be done without any privacy risks. We only use four different decay functions to show how different decay schedulers affect the model's accuracy and privacy without leading to privacy risks.

The noise decay scheduler involves adjustable factors such as the drop rate, initial noise multiplier, and epoch drop rate. The performance of the DP model depends on the optimization of these factors while avoiding inadvertent privacy breaches. To do this, Chaudhuri et al. (2011) has introduced a technique for achieving differentially private hyperparameter optimization. This method suggests splitting the data set into equivalent $k + 1$ segments. Subsequently, the $k$ models are trained using $k$ distinct schedules, each on a separate data segment. The performance of the model is assessed by counting the number of incorrect predictions for each model, represented as $z_i$ (where $1 \leq i \leq k$), in the remaining data segment. The Exponential Mechanism McSherry & Talwar (2007) is then used. This mechanism selects and produces a potential solution with a likelihood proportional to $exp(\frac{-z_i}{2})$. We leave the differentially private hyperparameter tuning to future work.

