# OpenReview forum: "Auto DP-SGD: Dual Improvements of Privacy and Accuracy via Automatic Clipping Threshold and Noise Multiplier Estimation"
_ICLR.cc/2024/Conference — Submitted to ICLR 2024_

### Official Review · Reviewer_Qfup · 2023-10-29

**Soundness:** 2 fair
**Presentation:** 2 fair
**Contribution:** 2 fair
**Rating:** 3
**Confidence:** 3

**Summary:**

The paper proposes a new privacy-preserving deep-learning training scheme, based on the existing DPSGD methods. The authors provide some theoretical explanation of the proposed method and verify the method on some classic datasets. The results show the proposed method has great performance on the utility.

**Strengths:**

1. The authors propose a detailed intro about the main analytic tools in DP and use them in the following part
2. The experiment parts are shown in detail with high reproductivity.

**Weaknesses:**

1. Presentation: 1) the authors need to clearly present some theoretical novelty and show some privacy preservation guarantee in the main context, together with the assumptions.
2) Some preliminary can be set in the Appendix.
3) some line spacing  is not well in the main context
2. Contents:
1) the related work needs to include more work in these two years. Much related DPSGD work is suitable here and I believe a table for the comparison might be better.
2) some magnitude about the privacy level should be shown, I just see some examples with the given values and am confused by the quantity. Some theoretical analysis should be given about how much improvement of the privacy level or utility level, is compared with the standard DPSGD.
3) More clarification about why the proposed method is important. I think it is just adjusting the variance level of DPSGD, it should influence the utility, but what we can lose in the privacy level should be presented. (Or what we derive)
4) More datasets should be considered since CIFAR is not enough for paper with the experiments consisting of the most parts. Large ones should be used.
5) Large $\epsilon$ seems to be meaningless in DP, but there are very large ones shown in the experiments.
6) how are parameters determined in the real treatment, like scaling factors, or variants. Some criteria should be shown.

**Questions:**

See the weaknesses parts.

---

### Official Review · Reviewer_CCmh · 2023-11-01

**Soundness:** 2 fair
**Presentation:** 2 fair
**Contribution:** 2 fair
**Rating:** 3
**Confidence:** 4

**Summary:**

The paper proposes Auto-DPSGD, a modified version of the DPSGD algorithm. Compared with the original DPSGD, Auto DP-SGD incorporates automatic clipping and noise multiplier scheduling.

**Strengths:**

Improving the privacy-utility trade-off of DPSGD is an interesting problem.

**Weaknesses:**

1. Auto DP-SGD does not satisfy differential privacy, because the noise that it injects into the gradient sum is a computed from the individual gradient norms, whereas individual gradient norms are private information.

To illustrate this issue, consider an extreme case where all records in the input dataset D have zero gradients. In that case, in Auto DP-SGD the distribution of the noisy gradient is degenerated to the point of zero (because s_t = 0 in Auto DP-SGD). Now consider neighboring dataset D’ that has one non-zero gradient. The noisy gradient sum distribution of D’ is a Gaussian (because s_t is non-zero). Comparing these two distributions of D and D’, it is apparently that they have unbounded divergence. This indicates a violation of differential privacy.

2. I am not convinced by the the authors’ claims that "CDP and zCDP accountants lack privacy amplification by sampling" and "the RDP accountant Abadi et al. (2016) overestimates privacy costs". Both CDP and RDP enjoy the same level of privacy amplification by subsampling, since they are both based on the Renyi divergence and can be converted to each other in many cases. Previous papers (e.g., see https://dl.acm.org/doi/pdf/10.1145/3219819.3220076) have also used CDP in DPSGD.

3. It is not a common practice to evaluate the performance of DP training algorithms on CIFAR10 and CIFAR100 *only* on pertained models. Please refer to https://arxiv.org/pdf/2204.13650.pdf for more comprehensive evaluation setups.

4. The paper misses existing work on automatic clipping and privacy budget scheduling. For example, see https://dl.acm.org/doi/pdf/10.1145/3219819.3220076 for one of the first papers that propose to use an adaptive privacy budget scheduling for DPSGD, and see https://arxiv.org/abs/2206.07136 for automatic clipping.

**Questions:**

Please refer to the weaknesses.

---

### Official Review · Reviewer_U13W · 2023-11-01

**Soundness:** 2 fair
**Presentation:** 2 fair
**Contribution:** 1 poor
**Rating:** 1
**Confidence:** 4

**Summary:**

This paper studies DP-SGD with a new way to estimate the clipping threshold automatically. It also proposes three ad-hoc noise multiplier decay mechanisms. Some experiments are provided on (mostly) toy models and toy datasets.

**Strengths:**

The proposed DP-SGD seems new and the experiments cover many aspects including learning rate, noise schedule, and task types.

**Weaknesses:**

1. First thing first, this paper is over the ICLR page limit (9 pages). This paper may be desk-rejected. Also some margins are violated by, for example, Table 2,3,4.

2. The contribution is not enough and not significant, e.g. the original content from this work only starts from page 5. Specifically, one of the "claimed" contribution is the proposal of three new noise multiplier decay mechanisms: Auto DP-T/S/E. However, Auto DP-S/E are useless according to Table 16 and thus not present in the main text. This is trivial novelty.

3. The proposed Algorithm 1 is computationally infeasible, essentially setting batch size as 1, and even so is memory prohibitive. Unlike modern DP-SGD, which takes batch size 64 for one back-propagation and can be as memory efficient as non-DP SGD (see "Differentially Private Optimization on Large Model at Small Cost"), Algorithm 1 needs 64 back-propagations and takes too long to train. As for the memory, the gradient list G from both Algorithm 1 and Algorithm 2 requires to store B*model_size parameter (B being batch size). This is impossible for large models with over 1B parameters, whereas state-of-the-art DP-SGD can already handle 100B parameters. In fact, I am shocked that the obvious computation inefficiency is not discussed in work.

4. Performance is too weak, datasets are too easy and comparison to old benchmarks is unfair. E.g. SOTA CIFAR100 at epsilon=2 is around 90%, where the authors got 79%; achieving >98% on MNIST can be traced back to 2020 (https://arxiv.org/pdf/2007.14191.pdf).

5. The name "Auto DP-SGD" is already used in other papers, possibly making the readers confused.

**Questions:**

See Weaknesses.

---

> ### Public Comment · ~Ali_Dadsetan1 · 2025-01-11
> **Request for Citation Regarding CIFAR-100 DP Performance**
>
> Dear Reviewer,
> I came across this insightful comments and greatly appreciate your analysis and contributions to the discussion.
>
> I noticed that you mentioned a 90% performance under Differential Privacy (DP) on the CIFAR-100 dataset. This is an impressive result, and I am keen to learn more about the methodology or paper achieving this performance. The best result I have been able to find so far is from the work by Bu, Z., Mao, J., & Xu, S. (2022), Scalable and efficient training of large convolutional neural networks with differential privacy, presented at NeurIPS 2022, which reports 88.4% accuracy on CIFAR-100 under DP constraints.
>
> Could you kindly provide a reference or citation for the 90% claim? It would be immensely helpful for my research and understanding of the state-of-the-art in this area.
>
> Thank you for your time and for enriching the discussion with your valuable insights.
>
> Best regards,

---

### Official Review · Reviewer_m4LY · 2023-11-04

**Soundness:** 2 fair
**Presentation:** 2 fair
**Contribution:** 2 fair
**Rating:** 5
**Confidence:** 5

**Summary:**

In this paper, various noise multiplier decay methods and different learning rate schedules were proposed and compared to identify a more effective approach for implementing DP-SGD, aiming to enhance accuracy.

**Strengths:**

This paper introduced and compared several noise multiplier decay methods and different learning rate schedules to identify a more effective approach for implementing DP-SGD, with the goal of enhancing accuracy

**Weaknesses:**

The effectiveness of both the adaptive clipping methods and the proposed learning rate schedules relies on experimental evidence. Consequently, conducting thorough experiments across a diverse range of machine learning models and datasets is imperative to assess the efficacy of these methods. Regrettably, this comprehensive evaluation is absent in this paper.

Furthermore, considering that hyperparameter tuning necessitates an additional DP budget, it becomes difficult to ascertain whether the effectiveness of the proposed methods stems from their inherent capabilities alone or if optimized hyperparameters have been employed to enhance their performance.

**Questions:**

According to my understand, DP protect the membership of input data, but how
does DP  protect "personally identifiable information (PII) in deep learning (DL) applications." as you claimed in the paper?

---

### Meta-Review · Area_Chair_TRAR · 2023-12-08

**Metareview:**

Reviewers were strongly opposed to the paper, and there were no rebuttal from the authors. In light of that, we recommend rejection, and encourage the paper to resubmit at another venue after taking care of the reviewer concerns.

**Justification For Why Not Higher Score:**

The reviewers were strongly negative.

**Justification For Why Not Lower Score:**

NA

---

### Decision · Program_Chairs · 2024-01-16

Reject